# Unifying Reinforcement Learning and Distillation via Distribution Matching for Video Generation

## Abstract

Reinforcement learning (RL) for aligning visual generative models faces dual challenges: (1) reward evaluation typically requires generation via extensive multi-step sampling (20-40 steps), and (2) existing GRPO-Based methods necessitate complex conversions to adapt ODE-based sampling in flow matching to the Markov Decision Process formulation. While distillation techniques enable few-step generation (e.g., 4 steps for a video), RL-after-distillation often leads to model collapse, whereas conventional workflows applying RL-before-distillation incur prohibitive computational costs. We address these limitations through a simple yet efficient unified framework that jointly optimizes alignment and distillation within a single stage. Inspired by Distribution Matching Distillation (DMD), our approach implements alignment directly via distribution matching (DM) through separately developed novel losses: DM-PairLoss (DPO-inspired) and DM-GroupLoss (GRPO-inspired). This methodology eliminates the need for reverse-SDE conversions while enabling direct reward evaluation from few-step generations. Comprehensive experiments on the Wan 2.1 text-to-video model demonstrate that our unified approach preserves distillation capabilities while achieving better human preference alignment, outperforming the raw base model, standalone distilled variant, and two-stage alignment-distillation alternatives on both VBench metrics and human evaluations. The synergistic optimization enhances both human preference alignment and distillation quality. We will release code and pretrained models to facilitate community research.

## 1 INTRODUCTION

Recent advances in visual generative modeling have achieved significant progress, primarily driven by diffusion models (Ho et al., 2020; Song et al., 2020a) and flow matching models (Lipman et al., 2022; Liu et al., 2022b). Both open-source research (Polyak et al., 2024; Kong et al., 2024; Wan et al., 2025) and proprietary commercial models (Kuaishou, 2024; Google DeepMind, 2024) can now generate high-quality images or videos. This progress has established a research paradigm mirroring large language models: after foundational base models are released, subsequent studies leverage these open-sourced base models to develop and validate new algorithms. In this work, we focus on two key research directions in image and video generation: 1) Distillation for diffusion (Yin et al., 2024b;a; Chadebec et al., 2025), developing techniques to reduce generation time while maintaining output quality, enabling few-step synthesis comparable to the original multi-step base model; 2) Post-training alignment, building upon the success of Reinforcement Learning from Human Feedback (RLHF) (Ouyang et al., 2022) in large language models, several recent studies (Liu et al., 2025b;a; Xue et al., 2025; Li et al., 2025) have demonstrated effective applications of Direct Preference Optimization (DPO) (Rafailov et al., 2023) and Group Relative Policy Optimization (GRPO) (Shao et al., 2024) to flow matching models for visual generation alignment.

Previous approaches for post-training alignment in generative modeling exhibit two significant limitations, as illustrated in Figure 1 (Left). First, the reward model requires complete image or video generation before scoring, whereas base models typically need extensive sampling steps (e.g., 20-40 steps) to produce outputs. Second, for flow matching models, the Ordinary Differential Equation (ODE)-based sampling of rectified flow conflicts with the Markov Decision Process (MDP) for-

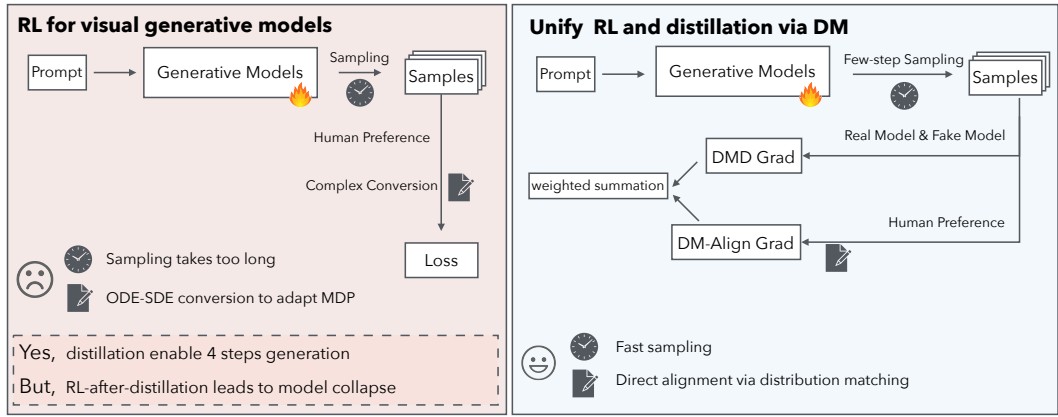

Figure 1: *Left:* Conventional RL suffers from slow sampling, ODE-SDE conversion complexity. *Right:* Our DM-based framework jointly optimizes alignment and distillation, enabling fast, high-quality generation with direct preference learning.

mulation. This incompatibility is typically resolved by unifying ODEs and Stochastic Differential Equations (SDEs) within an SDE framework, then deriving $\pi(a_t|s_t)$ through the corresponding reverse-time SDE (Song et al., 2020b; Albergo et al., 2023). Recent work has recognized the first issue: DanceGRPO (Xue et al., 2025) and Flow-GRPO (Liu et al., 2025a) reduce sampling steps and generation resolution without significantly compromising performance, while MixGRPO (Li et al., 2025) enhances sampling efficiency via SDE-ODE integration. Meanwhile, distilled models can generate outputs in few steps (Yin et al., 2024b;a; Luo et al., 2025) (e.g., 4 steps for a video). Although applying RL after distillation seems straightforward, fine-tuning such distilled models with RL losses frequently causes model collapse (manifesting as blurred outputs).[1] Consequently, the prevailing practice conducts RL before distillation (Chen et al., 2025). This approach suffers from the aforementioned long generation time during RL, compelling practitioners to use low-resolution or under-sampled videos that yield suboptimal results. Furthermore, the RL and distillation stages are not well-aligned.

To address these issues, we propose a simple yet efficient framework as shown in Fig. 1 (Right) that jointly optimizes alignment and distillation within a single unified stage. Inspired by Distribution Matching Distillation (DMD)-series works (Yin et al., 2024b;a), we implement alignment directly through distribution matching (DM). Specifically, we design specialized DM-PairLoss (DPO-inspired) and DM-GroupLoss (GRPO-inspired) that operate within this DM framework, enabling direct human preference alignment via distribution matching. These novel loss formulations constitute the alignment component of our unified optimization approach. Experimental results demonstrate that our unified approach achieves synergistic optimization by integrating both distillation and alignment stages. Crucially, our DM-based alignment eliminates the need for reverse-SDE conversions, enabling direct preference alignment and reward evaluation within the few-step generation. Implemented on the Wan 2.1 text-to-video (T2V-1.3B) base model (Wan et al., 2025) with VideoAlign (Liu et al., 2025b) as reward model, our method substantially outperforms all strong baselines—including the raw base model, standalone DMD distillation, DanceGRPO, flowDPO, and other two-stage baselines. This is demonstrated by significant gains of +4.58 and +6.55 in VBench average score for the Pair and Group variants, alongside a large net preference gain in human evaluations (+48% and +30% over base raw model, respectively), solidly validating the synergistic benefits of unified optimization.

To summarize, our key contributions are the following: (1) we propose a simple yet efficient framework that jointly optimizes alignment and distillation within a single unified stage, enabling few-step generation during alignment sampling and harmonizing optimization objectives across both stages to achieve superior results; (2) inspired by distribution matching (DM), we develop novel DM-PairLoss and DM-GroupLoss methodologies that directly optimize toward higher-reward distributions, achieving human preference alignment without the reverse-SDE conversions required in prior

---

[1]Detailed experimental results are provided in the Appendix C.1.

works; (3) through comprehensive experiments on the Wan 2.1 T2V-1.3B model, we demonstrate the mutual enhancement between distillation and alignment objectives, maintaining distillation quality while consistently improving human preference alignment.

## 2 RELATED WORK

### 2.1 DIFFUSION DISTILLATION

Image and video generation predominantly leverages diffusion models (Ho et al., 2020; Song et al., 2020a) or flow matching models (Lipman et al., 2022; Liu et al., 2022b), both requiring multi-step generation that introduces computational inefficiency and inference latency. Reducing sampling steps while preserving output quality thus remains a fundamental challenge. Early efforts (Liu et al., 2022a; Zhao et al., 2023; Karras et al., 2022) developed accelerated diffusion samplers to minimize step counts in pre-trained models. A straightforward idea, referred to as *trajectory distillation*, trains student networks to directly learn the teacher model's multi-step generation outputs. Progressive distillation (Salimans & Ho, 2022) implements this concept by gradually learning shortened generation trajectories through iterative refinement. Consistency distillation (Song et al., 2023b; Luo et al., 2023) enforces self-consistency across the PF-ODE trajectory, enabling student models to directly predict clean outputs from any intermediate noisy state. Alternatively, *distribution distillation* explicitly matches the teacher's output distribution. Adversarial-based methods (Mao et al., 2025; Lin et al., 2025) employ GAN-like frameworks that train discriminators and generators for direct few-step synthesis. Distribution Matching Distillation (DMD) (Yin et al., 2024b) and its follow-up works (Yin et al., 2024a; Shao et al., 2025) alternately optimize student models and fake models to minimize KL divergence between distributions. Recent efforts (Luo et al., 2025; Sun et al., 2025) integrate trajectory and distribution distillation techniques to achieve improved distillation performance.

Initial efforts focused on image distillation; while most image distillation algorithms transfer directly to video generation, few preserve performance on video tasks. Recent work addresses video distillation challenges involving larger models and more complex distribution matching. Prevailing approaches primarily employ GAN-based distribution matching frameworks. APT (Lin et al., 2025) achieves one-step generation via adversarial distribution matching, while POSE (Cheng et al., 2025) conducts DMD initialization before adversarial refinement.

Our work builds upon DMD (Yin et al., 2024b) and DMD2 (Yin et al., 2024a), initially designed for image distillation but later validated by the community as highly effective for video tasks with more streamlined implementation (LightX2V, 2025). Inspired by the DMD framework, multiple recent works (Luo et al., 2025; Shao et al., 2025; Sun et al., 2025; Gu et al., 2025) have proposed enhancements primarily focused on improving distillation efficiency. However, we uniquely exploit few-step video generation to enable efficient alignment training within a unified stage. Specifically, we integrate distillation and alignment through novel DM-based DPO-like and GRPO-like losses, enabling mutual enhancement where both objectives synergistically improve human preference alignment.

### 2.2 ALIGNING DIFFUSION MODELS AND FLOW MATCHING MODELS

The success of Reinforcement Learning from Human Feedback (RLHF) (Ouyang et al., 2022) in large language models has motivated its adaptation to diffusion and flow matching models. Current methods include training-free alignment (Yeh et al., 2024; Tang et al., 2024; Song et al., 2023a) that injects preference signals during sampling without model training. Beyond these, a significant line of work directly adapts RL algorithms to visual generation models. This includes: (i) DPO-based methods applied to image/video generation (Rafailov et al., 2023; Wallace et al., 2024; Yang et al., 2024; Liang et al., 2024; Zhang et al., 2025; Liu et al., 2025b), requiring positive/negative sample pairs with human preference annotations; and (ii) PPO-based methods (Schulman et al., 2017; Black et al., 2023; Fan et al., 2023; Liu et al., 2025a; Xue et al., 2025; Li et al., 2025) that derive preference signals from reward models without requiring paired data.

Although RL algorithms for image generation are theoretically transferable to video tasks, practical implementations encounter training instability and convergence difficulties. To address video-specific alignment, Flow-DPO (Liu et al., 2025b) proposes a dedicated video reward model and applies DPO to video generation models. DanceGRPO (Xue et al., 2025) achieves competitive results

in both image and video generation by applying GRPO within a unified SDE framework. However, this approach requires extensive sampling steps (typically 20-40) to compute losses at multiple trajectory points, incurring non-trivial computational overhead. Concurrent work MixGRPO (Li et al., 2025) enhances sampling efficiency through SDE-ODE alternation, though primarily focusing on image generation. In contrast, we adopt a novel DM perspective by designing GRPO-like and DPO-like losses that directly optimize distributions toward human-preferred outputs (higher-reward or positive samples). Crucially, we focus on unifying RL alignment and distillation—traditionally separate stages—within a single DMD-based optimization process, enabling direct preference optimization with enhanced efficiency and stability.

# 3 METHOD

In Section 3.1 we introduce Distribution Matching Distillation (DMD). In Section 3.2 we present our proposed DM-PairLoss(DPO-inspired) and DM-GroupLoss(GRPO-inspired) for alignment. In Section 3.3 we describe our algorithm framework that unifies RL and distillation into a single stage, along with a brief analysis of our implementation.

## 3.1 REVIEW OF DISTRIBUTION MATCHING DISTILLATION (DMD)

The DMD series of works (Yin et al., 2024b;a; 2025; Huang et al., 2025) have established the DMD framework. Our approach builds directly upon this foundation while preserving its most concise formulation without additional modifications. The core objective minimizes the Kullback-Leibler (KL) divergence between the distilled model's output distribution ($p_{\text{fake}}$) and the original teacher model's distribution ($p_{\text{real}}$):

$$\mathcal{L}_{\text{DMD}} = D_{\text{KL}}(p_{\text{fake}} \parallel p_{\text{real}}) \tag{1}$$

$$= \mathbb{E}_{x \sim p_{\text{fake}}} \left[ \log \frac{p_{\text{fake}}(x)}{p_{\text{real}}(x)} \right] = \mathbb{E}_{\substack{z \sim \mathcal{N}(0;\mathbf{I}) \\ x = G_\theta(z)}} \left[ -(\log p_{\text{real}}(x) - \log p_{\text{fake}}(x)) \right]. \tag{2}$$

Since directly estimating probability densities for Equation 2 is intractable, DMD trains the generator via gradient descent. This requires computing the gradient with respect to parameters $\theta$, approximated by perturbing data distributions with Gaussian noise to create overlapping "blurred" distributions:

$$\nabla_\theta \mathcal{L}_{\text{DMD}} \approx -\mathbb{E}_t \int \left[ \mathbf{s}_{\text{real}}\Big( F(G_\theta(z), t), t \Big) - \mathbf{s}_{\text{fake}}\Big( F(G_\theta(z), t), t \Big) \right] \frac{dG_\theta(z)}{d\theta} dz. \tag{3}$$

Here $z \sim \mathcal{N}(0, \mathbf{I})$ is Gaussian noise input, $F$ represents the forward diffusion process at noise level $t$ (sampled randomly across timesteps), and $\mathbf{s}_{\text{real}}(x) = \nabla_x \log p_{\text{real}}(x)$, $\mathbf{s}_{\text{fake}}(x) = \nabla_x \log p_{\text{fake}}(x)$ denote distribution scores. As shown in Fig. 2, our implementation maintains the original DMD framework: the real model serves as a fixed teacher, while the fake model approximates the generator's output distribution via denoising loss. We initialize the fake model from the pretrained model, updating parameters $\phi$ during training by minimizing the standard denoising objective:

$$\mathcal{L}_\phi^{\text{denoise}} = \|\mu_\phi^{\text{fake}}(\mathbf{x}_t, t) - \mathbf{x}_0\|_2^2, \tag{4}$$

where $\mathbf{x}_0$ represents the target sample. To enhance stability, we employ alternating training cycles (5 fake model updates per generator update). Our framework preserves the essential components of DMD2 while omitting auxiliary losses.

## 3.2 DM-ALIGN LOSSES

From a distribution matching (DM) perspective, a straightforward approach would directly align the generator's output distribution toward human-preferred distributions. We can intuitively design an alignment loss:

$$\mathcal{L}_{\text{align}} = D_{\text{KL}}(p_{\text{gen}} \parallel p_{\text{pref}}) = \mathbb{E}_{x \sim p_{\text{gen}}} \left[ \log \frac{p_{\text{gen}}(x)}{p_{\text{pref}}(x)} \right]. \tag{5}$$

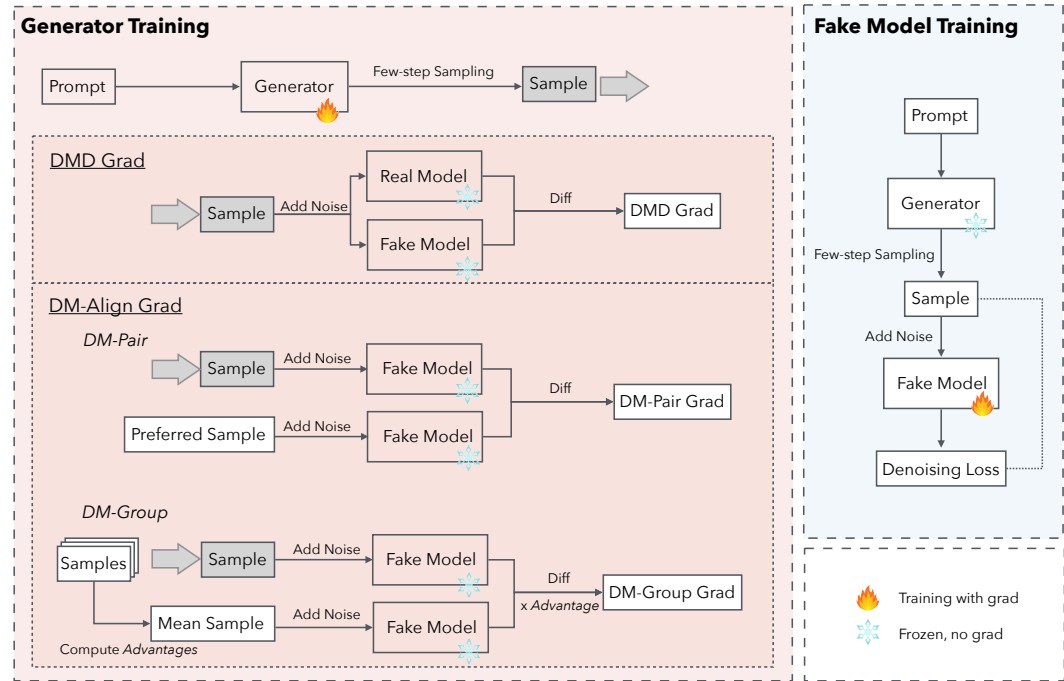

Figure 2: **Overview.** Unified training framework integrating DMD and DM-Align. *Left:* Generator produces samples via few-step sampling, updated by DMD Grad and DM-Align Grad. *Right:* Fake model approximates generator distribution via denoising loss on noisy few-step samples.

However, we cannot access a reliable score model ($\nabla_x \log p_{\text{pref}}(x)$) for human-preferred distributions, preventing direct optimization of this KL divergence. Conventional RL methods acquire preference information through either: (1) pre-annotated human preference data (DPO-based), or (2) reward models that score multiple generated samples (GRPO-based). Building upon the gradient formulation in Equation 3, we develop a sample-based distribution matching approach for alignment. Within the DMD framework, the fake model provides a stable distribution estimate. We thus optimize:

$$\nabla_\theta \mathcal{L}_{\text{align}} \approx -\mathbb{E}_t \int \left[ \mathbf{s}_{\text{fake}}\Big( F(x_+, t), t \Big) - \mathbf{s}_{\text{fake}}\Big( F(G_\theta(z), t), t \Big) \right] \frac{dG_\theta}{d\theta} dz, \quad (6)$$

where $x_+$ denotes human-preferred samples. Intuitively, this drives the generator toward higher-reward outputs. We can derive this gradient through reasonable approximations starting from the Bradley-Terry preference model (Bradley & Terry, 1952), with detailed derivation provided in Appendix A. The fake model is chosen for estimation stability due to its alternating update pattern (5 updates per generator step) and consistent distribution approximation during training.

**DM-PairLoss.** For DPO-like alignment, we design an online loss inspired by Flow-GRPO's online-DPO approach (Liu et al., 2025a). This practical approach requires only human-preferred positive samples (typically SFT data), which are readily available, while utilizing generator outputs as negative samples. We directly substitute the corresponding samples in Equation 6 to obtain the gradient for DM-PairLoss.

**DM-GroupLoss.** Similarly inspired by GRPO, we adopt a group-based approach: for each input, we sample multiple outputs from the generator, forming a sample group. Within this group, we compute rewards and corresponding advantages using a reward model. We consider the average of the top-$n$ samples as an "average estimate" of human-preferred outputs and apply advantage as the scaling factor for gradient updates. This yields the gradient for DM-GroupLoss:

$$\nabla_\theta \mathcal{L}_{\text{DM-Group}} \approx -\mathbb{E}_t \int A \left[ \mathbf{s}_{\text{fake}}\Big( F(\bar{x}_+, t), t \Big) - \mathbf{s}_{\text{fake}}\Big( F(G_\theta(z), t), t \Big) \right] \frac{dG_\theta}{d\theta} dz, \quad (7)$$

---

**Algorithm 1** DMD Training with Human Preference Alignment

---

**Require:** Pretrained real model $\mu_{\text{real}}$, conditional input dataset $\mathcal{D} = \{\mathbf{c}\}$, preference data $\mathcal{D}_{\text{pref}}$.
**Require:** Few-step denoising timesteps $\mathcal{T} = \{t_1, \ldots, t_Q\}$, update ratio $N$, group size $k$.
1: $G_\theta \leftarrow \texttt{copyWeights}(\mu_{\text{real}})$; $\mu_{\text{fake}} \leftarrow \texttt{copyWeights}(\mu_{\text{real}})$
2: **while** training **do**
3:      Sample batch $\mathbf{z} \sim \mathcal{N}(0, I)^B$ and $\mathbf{c} \sim \mathcal{D}$     ▷ Generate samples with intermediate timesteps
4:      Sample $t \sim \text{Uniform}(\mathcal{T})$
5:      $\mathbf{x}, \mathbf{x}_{\text{final}} \leftarrow \texttt{generateSamples}(G_\theta, \mathbf{z}, \mathbf{c}, t)$
6:      $\nabla\mathcal{L}_{\text{DMD}} \leftarrow \texttt{DMDGradient}(\mu_{\text{real}}, \mu_{\text{fake}}, \mathbf{x})$                      ▷ Eq. 3
7:      **if** using DM-PairLoss **then**
8:          $\mathbf{x}_+ \leftarrow \texttt{sample}(\mathcal{D}_{\text{pref}}, \mathbf{c})$
9:          $\nabla\mathcal{L}_{\text{align}} \leftarrow \texttt{DM-PairGradient}(\mu_{\text{fake}}, \mathbf{x}_{\text{final}}, \mathbf{x}_+)$          ▷ Eq. 6
10:     **else if** using DM-GroupLoss **then**
11:         $\mathbf{x}_{\text{group}} \leftarrow \texttt{groupSamples}(\mathbf{x}_{\text{final}}, k)$
12:         $\nabla\mathcal{L}_{\text{align}} \leftarrow \texttt{DM-GroupGradient}(\mu_{\text{fake}}, \mathbf{x}_{\text{final}}, \mathbf{x}_{\text{group}})$      ▷ Eq. 7
13:     **end if**
14:     $\nabla\mathcal{L}_G \leftarrow \lambda_{\text{DMD}}\nabla\mathcal{L}_{\text{DMD}} + \lambda_{\text{align}}\nabla\mathcal{L}_{\text{align}}$          ▷ Combined gradient update
15:     $G_\theta \leftarrow \texttt{update}(G_\theta, \nabla\mathcal{L}_G)$
16:     **for** $i = 1$ **to** $N$ **do**                                ▷ Update fake model $N$ times
17:         Sample $t \sim \mathcal{U}(0, 1)$ and $\epsilon \sim \mathcal{N}(0, I)$
18:         $\mathbf{x}_t \leftarrow \alpha_t * \texttt{stopgrad}(\mathbf{x}) + \sigma_t\epsilon$
19:         $\mathcal{L}_{\text{denoise}} \leftarrow \texttt{denoisingLoss}(\mu_{\text{fake}}(\mathbf{x}_t, t), \texttt{stopgrad}(\mathbf{x}))$        ▷ Eq. 4
20:         $\mu_{\text{fake}} \leftarrow \texttt{update}(\mu_{\text{fake}}, \nabla\mathcal{L}_{\text{denoise}})$
21:     **end for**
22: **end while**

---

where $\bar{x}_+$ represents the average of high-advantage samples and $A$ denotes the advantage. Intuitively, for low-advantage samples, we shift distributions toward preferred outputs; for high-advantage samples, we update toward the specific sampled outputs from the average estimate. We observe that for a given input, rewards exhibit limited variation across multiple generated samples, thus we forgo advantage clipping. Instead, we apply gradient normalization to maintain scale compatibility with distillation gradients.

### 3.3 Unified Reinforcement Learning and Distillation

As illustrated in Fig. 2, we unify the RL and distillation stages by directly combining the DMD loss and DM-Align loss through weighted summation within the DMD framework:

$$\mathcal{L} = \lambda_{\text{align}}\mathcal{L}_{\text{align}} + \lambda_{\text{DMD}}\mathcal{L}_{\text{DMD}}, \tag{8}$$

where $\lambda_{\text{align}}$ and $\lambda_{\text{DMD}}$ are weighting coefficients. The complete training procedure is detailed in Algorithm 1. During training, we maintain stability through alternating updates of the generator and fake model, preserving DMD2's concise structure. The DMD loss is computed at a randomly sampled timestep $t \sim \mathcal{T}$ from the few-step denoising schedule. Simultaneously, the generator produces final outputs $\mathbf{x}_{\text{final}}$ by completing the full few-step denoising process. These $\mathbf{x}_{\text{final}}$ outputs provide more accurate estimates for human preference alignment: they serve as high-quality reference samples for pairwise comparison in DM-PairLoss or as reliable inputs for reward scoring in DM-GroupLoss. The DMD loss enables efficient few-step sample generation, while the DM-Align loss directly optimizes the generator to align its output distribution with human preferences using these samples.

The weighted combination of multiple objectives is well-established, as demonstrated in both the original DMD series (Yin et al., 2024b;a) and other distillation methods (Chadebec et al., 2025; Shao et al., 2025). However, while prior works focused on enhancing distillation quality, our approach prioritizes alignment with human preferences. The recent TDM method (Luo et al., 2025) suggests accelerating fake model training via importance sampling. Although incorporating additional losses may improve generation quality and alternative frameworks could enhance training efficiency, we preserve DMD2's core workflow for simplicity and leave these enhancements for future exploration.

Beyond KL minimization, we provide intuitive interpretations and visualizations of our DM loss in Appendix B.

# 4 EXPERIMENTS

## 4.1 EXPERIMENT SETUP

We employ Wan 2.1-T2V-1.3B (Wan et al., 2025) as our base model, which is a flow matching-based model that generates 5-second videos at 16 FPS with a resolution of 832 × 480. All training and evaluation are conducted at this resolution. For the DPO-based method, preferred samples are center-cropped to a matching aspect ratio. We term the pair-based and group-based variants of our method (Section 3) as **DM-Align (Pair)** and **DM-Align (Group)**, respectively.

**For DM-Align (Pair)**, we adopt the filtered ConsistID dataset (Yuan et al., 2025). We retain videos longer than 5 seconds from the DanceGRPO-filtered subset, resulting in roughly 6K samples with corresponding text prompts. This dataset comprises real videos, which inherently exhibit higher visual and motion quality than generated content, making it suitable for preference alignment. Baselines include the raw model, standalone DMD2, standalone flowDPO, and a two-stage flowDPO+DMD2 pipeline. We implement an online sampling version of flowDPO, which was proposed in FlowGRPO (Liu et al., 2025a) and has been demonstrated to achieve better performance.

**For DM-Align (Group)**, we use the filtered VidProM dataset (Wang & Yang, 2024) following previous work (Xue et al., 2025). This dataset contains approximately 100K diverse text prompts after filtering, though we only utilize 20K during training. Our approach relies solely on textual prompts. However, the VideoAlign reward model (Liu et al., 2025b) exhibits certain limitations when evaluating motion quality (MQ) and visual quality (VQ). Specifically, it tends to assign erroneously high scores to degenerative content such as pure white noise or flickering frames. This issue makes both DanceGRPO and our method susceptible to training collapse. Therefore, we primarily focus on textual alignment (TA) for this dataset. Comparative baselines include the raw base model, standalone DMD2, standalone DanceGRPO, and a two-stage DanceGRPO+DMD2 pipeline.

**For training**, all experiments are conducted on 32 H100 GPUs. Hyperparameters largely follow the DMD2 setup for fair comparison. In our combined loss function (Eq. 8), both $\lambda_{\text{align}}$ and $\lambda_{\text{DMD}}$ are set to 0.5. For other baselines, we adhere to the settings reported in their original papers to the extent possible. Both DMD2 and flowDPO use Exponential Moving Average (EMA) for training stability, while our method does not, as we observed negligible performance difference. DM-Align (Group) and DMD2 converge stably within 500 steps (∼5 hours). DM-Align (Pair) is trained for an additional 500 steps to ensure convergence. FlowDPO is also trained for 500 steps, while each step of DanceGRPO takes approximately 20 minutes; we train it for 100 steps.

**For evaluation**, we adopt VBench (Huang et al., 2024) as our automated evaluation metric, which includes 6 key metrics across 3 primary dimensions: temporal consistency, motion quality, and visual quality. For important comparisons, we conduct a human evaluation using the GSB (Good, Same, Bad) protocol. We curated a test set of 640 samples for both automated and human evaluations. For human evaluation, we select the checkpoints that achieve the highest VBench scores.

## 4.2 MAIN EXPERIMENTS

The experimental results on VBench are summarized in Table 1, where NFE denotes the Number of Function Evaluations. DM-Align (Pair) outperforms all baselines, achieving the highest average score (82.78) with only 4 NFE. Both our method and Flow-DPO significantly improve Motion Quality, but Flow-DPO shows limited gains in Visual Quality. While the two-stage Flow-DPO + DMD2 pipeline can partially recover Visual Quality, it diminishes Motion Quality benefits. In contrast, our single-stage optimization stably improves both aspects simultaneously.

For DM-Align (Group), our method achieves the highest average score (84.40), showing substantial improvements over other baselines. However, similar to the pair variant, the two-stage DanceGRPO + DMD2 pipeline suffers from performance degradation (e.g., 14.93 drop in Dynamic Degree) after distillation. Our approach eliminates multi-step sampling and alignment without timestep averaging, reducing training time by approximately 6× compared to DanceGRPO. The Textual Alignment

Table 1: Main results on VBench comprehensive evaluation. The best results are in **bold**. The upper section presents results on the ConsistID dataset using DM-Align (Pair), while the lower section shows results on the VidProM dataset using DM-Align (Group).

| Method | NFE | Average Score | Temporal Consistency | | Motion Quality | | Visual Quality | |
| --- | --- | --- | --- | --- | --- | --- | --- | --- |
| | | | Subject Consistency | Background Consistency | Motion Smoothness | Dynamic Degree | Aesthetic Quality | Imaging Quality |
| *DM-Align (Pair) on ConsistID Dataset* | | | | | | | | |
| Wan-T2V-1.3B (raw model) | 100 | 78.20 | 96.57 | 94.46 | 99.01 | 50.78 | 56.09 | 72.29 |
| DMD2 | 4 | 79.68 | 98.03 | 95.78 | 99.10 | 52.81 | 57.64 | 74.73 |
| Flow-DPO | 100 | 81.54 | 98.10 | 95.56 | 99.25 | 67.13 | 56.32 | 72.91 |
| Flow-DPO + DMD2 | 4 | 79.89 | 98.14 | 96.01 | 99.18 | 53.43 | 58.26 | 74.35 |
| **DM-Align (Pair) (Ours)** | 4 | **82.78** | **98.80** | 95.76 | **99.35** | 66.25 | **60.89** | **75.61** |
| *DM-Align (Group) on VidProM Dataset* | | | | | | | | |
| Wan-T2V-1.3B (raw model) | 100 | 77.85 | 93.93 | 94.63 | 98.63 | 56.40 | 57.33 | 66.20 |
| DMD2 | 4 | 78.88 | 95.53 | 95.22 | 98.60 | 51.64 | **63.02** | 69.30 |
| DanceGRPO | 100 | 82.76 | 96.04 | 95.86 | 98.33 | 75.78 | 60.44 | 70.15 |
| DanceGRPO + DMD2 | 4 | 80.54 | 96.60 | 95.89 | 98.43 | 60.85 | 60.71 | 70.81 |
| **DM-Align (Group) (Ours)** | 4 | **84.40** | **97.13** | **96.49** | **98.99** | **80.19** | 62.17 | **71.45** |

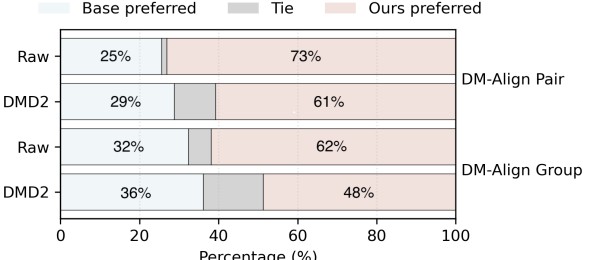

Figure 3: Human evaluation results.

Table 2: Textual Alignment (TA) scores evaluated by VideoAlign reward model on the VidProM dataset.

| Method | TA Score |
| --- | --- |
| Wan-T2V-1.3B (raw model) | 0.69 |
| DMD2 | 0.75 |
| DanceGRPO | 1.01 |
| DanceGRPO + DMD2 | 1.05 |
| **DM-Align (Group) (Ours)** | **1.65** |

scores (Table 2) further confirm our method provides stronger guidance than all baselines (1.65 vs. next best 1.05).

Human evaluation results (Fig. 3) indicate that both variants of our method significantly outperform the raw model and DMD2 in human preference. DM-Align (Pair) shows more pronounced improvement (+48 and +32 percentage points) as it leverages real human preference data, providing more accurate gradient directions for alignment. Although DM-Align (Group) relies on multiple sampling and reward model estimations for preference-guided gradients, it still delivers substantial performance gains (+30 and +12 percentage points). Appendix D provides several comparative examples of generated results. Overall, DMD2 effectively distills the 100-NFE generation process into a more efficient 4-NFE few-step generation, demonstrating its inherent stability. Our single-stage approach successfully integrates alignment into this framework without compromising its distillation capabilities.

## 4.3 VALIDATING THE DESIGN

In this section, we present a deeper analysis of our proposed algorithm. Our goal is to unify RL and distillation within a single stage under the DMD framework through our novel DM-Align loss. A straightforward alternative would be to directly incorporate the RL loss from DanceGRPO or Flow-DPO into the DMD optimization process. To this end, we developed a variant for DM-Align (Group) that simply adds the DanceGRPO loss alongside the standard DMD loss. Using an additional 128 prompts during training, we monitored the reward progression, as illustrated in Fig. 4.

Our results show that our method achieves a significant improvement in Textual Alignment (TA) over standalone DMD. In contrast, the DanceGRPO variant performs nearly identically to DMD. A similar phenomenon occurs with a Flow-DPO variant, where combining it with DMD yields results almost indistinguishable from DMD alone (e.g., a difference in VBench average score of less than 0.03). We attribute this limitation to the overly conservative gradient updates characteristic of these standalone RL losses. This conservatism stems from the instability of applying RL to

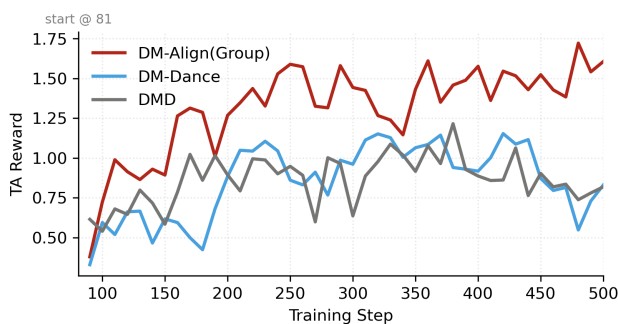

Figure 4: Reward progression during the training process for different methods.

generative models, where aggressive updates can lead to rapid performance collapse. Consequently, their corresponding loss terms are designed to be very conservative to maintain stability. While we acknowledge that more update steps with these methods might eventually lead to improvement, our DM-Align loss is inherently more compatible with the DMD framework, leading to superior efficiency and effectiveness. A discussion of other potential designs and variants is provided in Appendix C.2.

## 5 CONCLUSION

In this paper, we unify video generation model distillation and RL into a single stage within the DMD framework through distribution matching—traditionally considered separate stages. Our approach eliminates complex conversions (e.g., adapting ODE-based flow matching sampling to MDP formulations) and avoids time-intensive multi-step generation for reward computation required by prior RL methods. We evaluate our approach on the Wan 2.1 T2V-1.3B model. Experimental results show that our method maintains the efficiency of distillation while achieving improved alignment with human preferences. It outperforms the original base model, individually distilled models, and conventional two-stage pipelines on both automated VBench metrics and human evaluation.

However, several limitations exist: First, our straightforward weighted summation of losses lacks theoretical convergence guarantees. Second, by relying on DMD, our method restricts distillation to DMD-based techniques, excluding other recent GAN-based alternatives (Mao et al., 2025; Lin et al., 2025; Cheng et al., 2025) that have been shown to achieve better distillation. We believe this work provides inspiration for future research in two directions: a more detailed theoretical analysis of DM-based losses (including convergence and generalization properties), and the exploration of how to unify RL and distillation within broader distribution matching frameworks. More broadly, it invites a higher-level perspective on distillation and RL as two commonly employed yet often separated stages in base model optimization.

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

## A    FROM BRADLEY-TERRY PREFERENCE TO DM-ALIGN GRADIENT

This appendix outlines the motivation and conceptual derivation behind the alignment gradient in Equation equation 6, which connects preference learning with distribution matching.

**Preference Model and Loss.**    We adopt the Bradley-Terry model to define the probability that a human-preferred sample $x_+$ is ranked higher than a generated sample $x = G_\theta(z)$, using the log-likelihood under the *fake* model's distribution $s_\theta(\cdot) = \log p_\theta(\cdot)$ as the scoring function:

$$P_\theta(x_+ \succ x) = \frac{\exp(s_\theta(x_+))}{\exp(s_\theta(x_+)) + \exp(s_\theta(x))}. \tag{9}$$

The associated loss for a pair $(x, x_+)$ is the negative log-likelihood:

$$\mathcal{L}_{\text{pair}} = -\log P_\theta(x_+ \succ x) = \log\left(1 + \frac{p_\theta(x)}{p_\theta(x_+)}\right). \tag{10}$$

**Motivation via an Approximate Objective.**    Directly optimizing $\mathcal{L}_{\text{pair}}$ is challenging. To derive a more tractable objective, we consider the scenario where the generator is initially unaligned, leading to a large quality gap: $p_\theta(x_+) \gg p_\theta(x)$. Under this *imbalance assumption* ($\varepsilon := p_\theta(x)/p_\theta(x_+) \ll 1$), a first-order Taylor expansion yields:

$$\mathcal{L}_{\text{pair}} = \log(1 + \varepsilon) \approx \varepsilon = \frac{p_\theta(x)}{p_\theta(x_+)}. \tag{11}$$

Up to an additive constant, this can be further approximated by:

$$\mathcal{L}_{\text{pair}} \approx -(\log p_\theta(x_+) - \log p_\theta(x)) = s_\theta(x) - s_\theta(x_+). \tag{12}$$

Taking expectations, we arrive at an approximate overall objective:

$$\mathcal{L}_{\text{align}}(\theta) \approx \mathbb{E}_{z,x_+}\left[s_\theta(x) - s_\theta(x_+)\right]. \tag{13}$$

Minimizing Eq. equation 13 simultaneously minimizes the likelihood of generated data while maximizing the likelihood of preferred data, effectively performing distribution matching between $p_{\text{gen}}$ and $p_{\text{pref}}$.

**From Objective to Practical Gradient.**    The exact gradient of $s_\theta(\tilde{x}) = \log p_\theta(\tilde{x})$ is intractable. However, the DMD framework (Yin et al., 2024b) provides a practical and stable approximation for its direction. Specifically, DMD uses the *fake* model's score function $\mathbf{s}_{\text{fake}}$ along the probability flow to estimate the gradient of the log-likelihood for any sample $\tilde{x}$.

Inspired by this, we approximate the gradient of our objective $\mathcal{L}_{\text{align}}$ by leveraging the fake model's score. This leads us to the practical gradient update rule used in our method:

$$\nabla_\theta \mathcal{L}_{\text{align}} \approx -\mathbb{E}_{t,z,x_+}\left[\left(\mathbf{s}_{\text{fake}}\big(F(x_+, t), t\big) - \mathbf{s}_{\text{fake}}\big(F(G_\theta(z), t), t\big)\right) \cdot \frac{dG_\theta}{d\theta}\right], \tag{14}$$

which is the form presented in Equation equation 6. This gradient intuitively pushes the generator towards regions of high density under the fake model's estimate of the preferred data distribution.

## B    INTERPRETATION OF THE DM-ALIGN LOSS VIA VISUALIZATION

This appendix provides visualizations of key intermediate results during DMD training, offering an intuitive perspective to interpret how DMD operates. Through these visual explanations, we aim to develop a simpler understanding of our proposed DM-Align loss and elucidate why it effectively aligns generative models with human preferences.

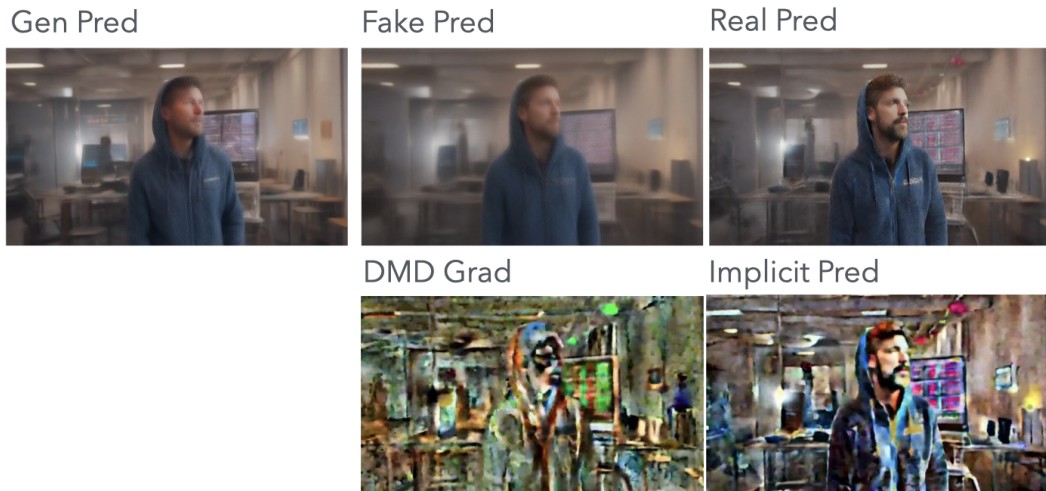

Figure 5: Visualization of DMD gradient and latent predictions. *Top row:* Predictions from the generator (Gen Pred), fake model (Fake Pred), and fixed real model (Real Pred). *Bottom row:* DMD gradient visualization (DMD Grad) and implicit target latent sample derived from gradient updates (Implicit Pred).

**Visualizing the DMD Grad** Since the DMD loss operates in the latent space, we visualize its core components (gradient direction, model predictions, and implicit update targets) by decoding latent vectors through a corresponding VAE decoder. As shown in Fig. 5, during DMD training, the initial few-step generations from the model are blurry, and the predictions from the fake model are similarly indistinct. However, the fixed real model provides a relatively clearer and more stable target. This creates a gradient direction that effectively points from a blurry state towards a sharper one, roughly outlining the contour of a higher-quality sample. By computing this gradient and converting it into a model update via MSE, we can visualize the implicit "target" latent sample (Implicit Pred). The discrepancy between the real and fake models provides an update signal that pushes the current generated latent towards a slightly clearer version, guiding the generator to produce higher-quality samples iteratively.

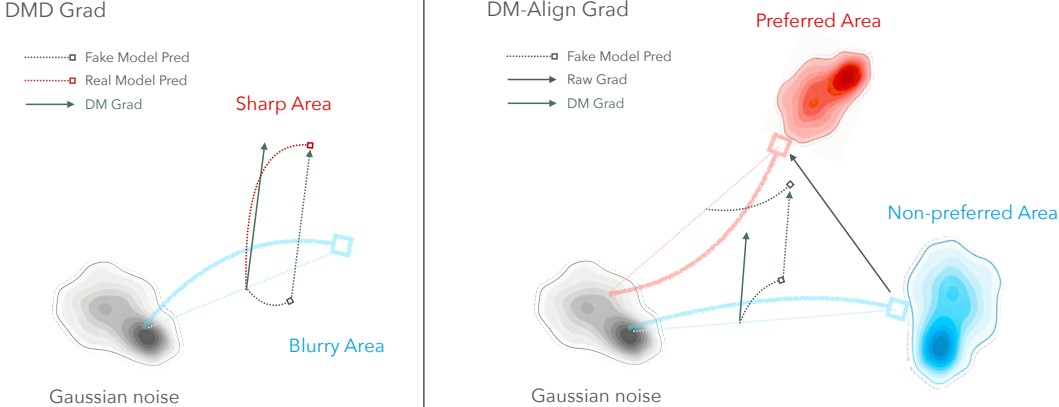

Figure 6: An intuitive visualization of the gradient directions. (Left) The DMD gradient. (Right) The DM-Align gradient. Both methods utilize a simpler, more attainable proximal target to guide each optimization step.

**Analogous Interpretation of DM-Align** From this perspective, our DM-Align gradient functions analogously. While standard DMD drives the update from a blurry generation towards a sharper one (a clarity advantage innate to the fixed real model), our DM-Align gradient drives the update

from a generated sample that is poorly aligned with human preferences towards one that is better aligned (an advantage conferred by the annotated preference data or reward model assessment). An intuitive visualization of this process is shown in Fig. 6. We conjecture that the core efficacy of the algorithm stems from its use of a simpler, more attainable proximal target for each update. Although theoretically, a more direct gradient estimate (e.g., towards the ground-truth sample or without noise scheduling) might seem preferable, our validation experiments (Appendix C.1) found that such approaches often lead to stable training loss but persistently blurry and low-quality generated content. In essence, both methods leverage the fake model's unified estimation and noise scheduling to establish a consistent optimization path. DMD leverages the additional fixed real model to guide this path towards incremental clarity, while our method leverages human preference signals (from pairwise comparisons or reward models) to guide this path towards incremental alignment with human preferences.

## C  ANALYSIS OF ALTERNATIVE APPROACHES

### C.1  TRAINING ON DISTILLED MODELS

Video generation models, after distillation, can produce high-quality videos in very few sampling steps. A seemingly straightforward approach is to perform RL training after the distillation process, leveraging this fast sampling to compute rewards efficiently. However, our experiments reveal that fine-tuning the distilled model in this manner leads to rapid degradation in generated video quality, manifesting as noticeable blurring artifacts and flickering motions, which ultimately progresses to a complete model collapse. This phenomenon is visually demonstrated in Fig. 7.

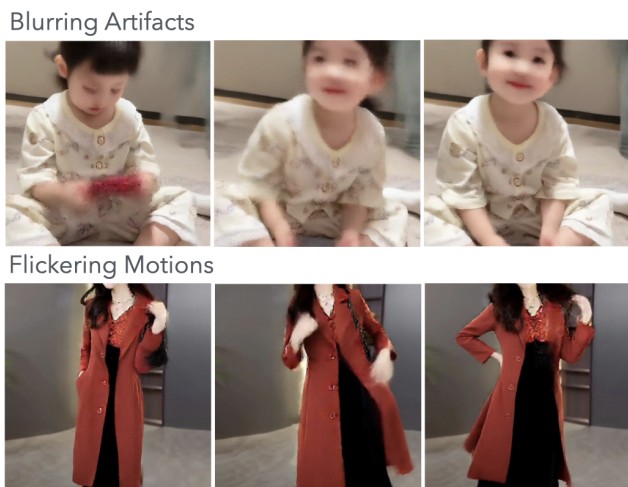

Figure 7: Visual examples of model collapse when fine-tuning a distilled model.

In practice, we found that any subsequent training (be it SFT or RL variants) on top of a distilled model is highly susceptible to collapse. Although the training loss might decrease, the model gradually degrades, producing blurred content and losing its few-step generation capability. We hypothesize that the distilled model's parameters become extremely sensitive; even minor adjustments can destabilize the output, making it exceedingly difficult to optimize for objectives beyond the original distillation goal.

### C.2  OTHER VARIANTS

Our DM-Align is designed based on the core structure of DMD2, introducing an alignment loss atop it. Naturally, numerous design choices and potential variants exist. Specifically, key considerations include whether to add noise before computing the alignment gradient, and whether to estimate this gradient using the generator, the (DMD) fake model, the real model, or another approximation.

We empirically explored many of these variants. While several achieved decent results, our final design in Sec. 3 was selected for its superior training stability while maintaining simplicity. We note that the gradient information in our Align loss—computed across different samples using the same fake model—and that in the DMD loss—computed across different fake and real models for the same sample—exhibit a complementary, almost dual-like, structural relationship. We hope our work inspires future research into better frameworks for unifying RL and distillation.

# D MORE VISUALIZATION RESULTS

In this section, we present additional visualization results of our proposed method, where the upper part of each figure shows results from the raw model, and the lower part presents results after applying DM-Align.

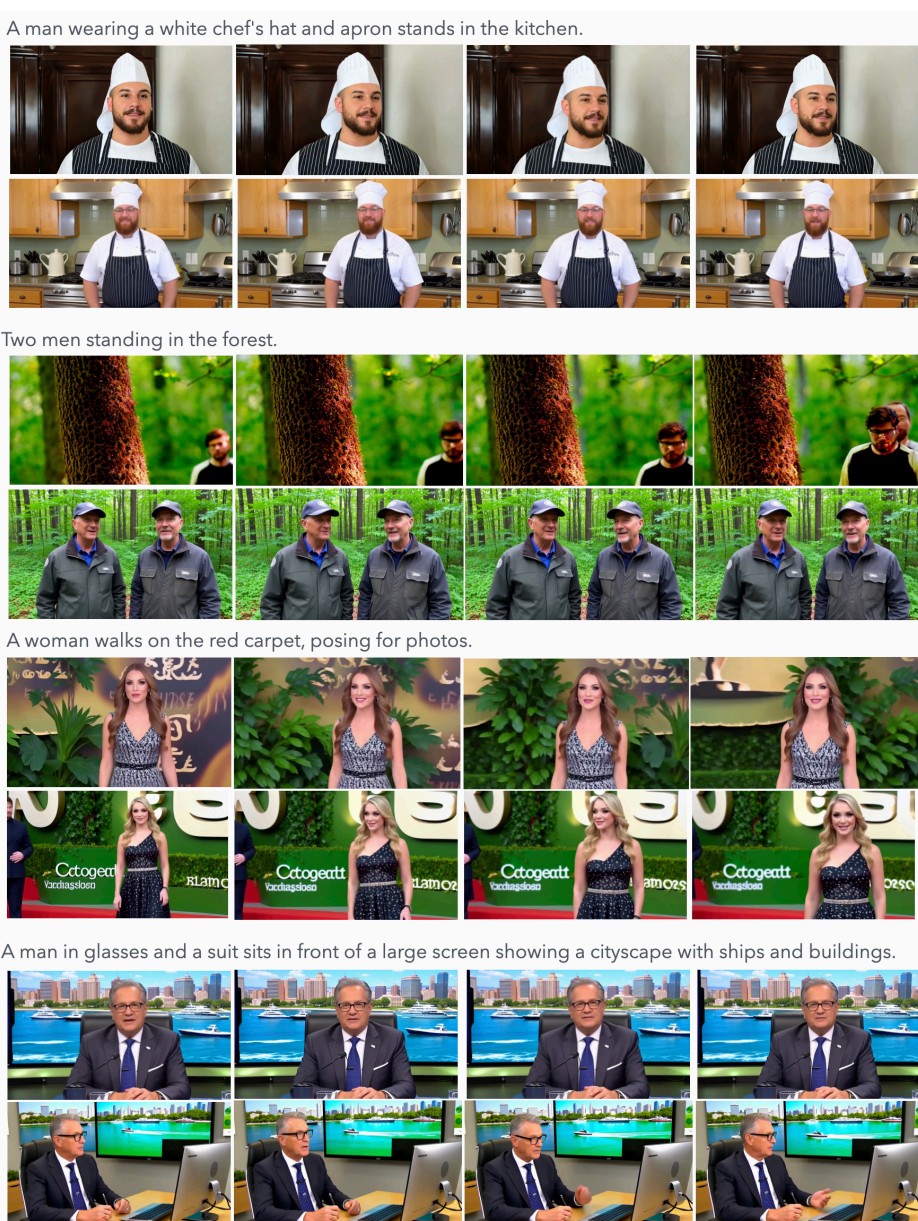

Figure 8: Visualization results of DM-Align (Pair). From top to bottom, compared with the base model, our improvements include: correctly generating the kitchen background, generating two persons with more natural movement (previously frozen in base model outputs) and improved yacht movement in the background (previously static in base model generations).

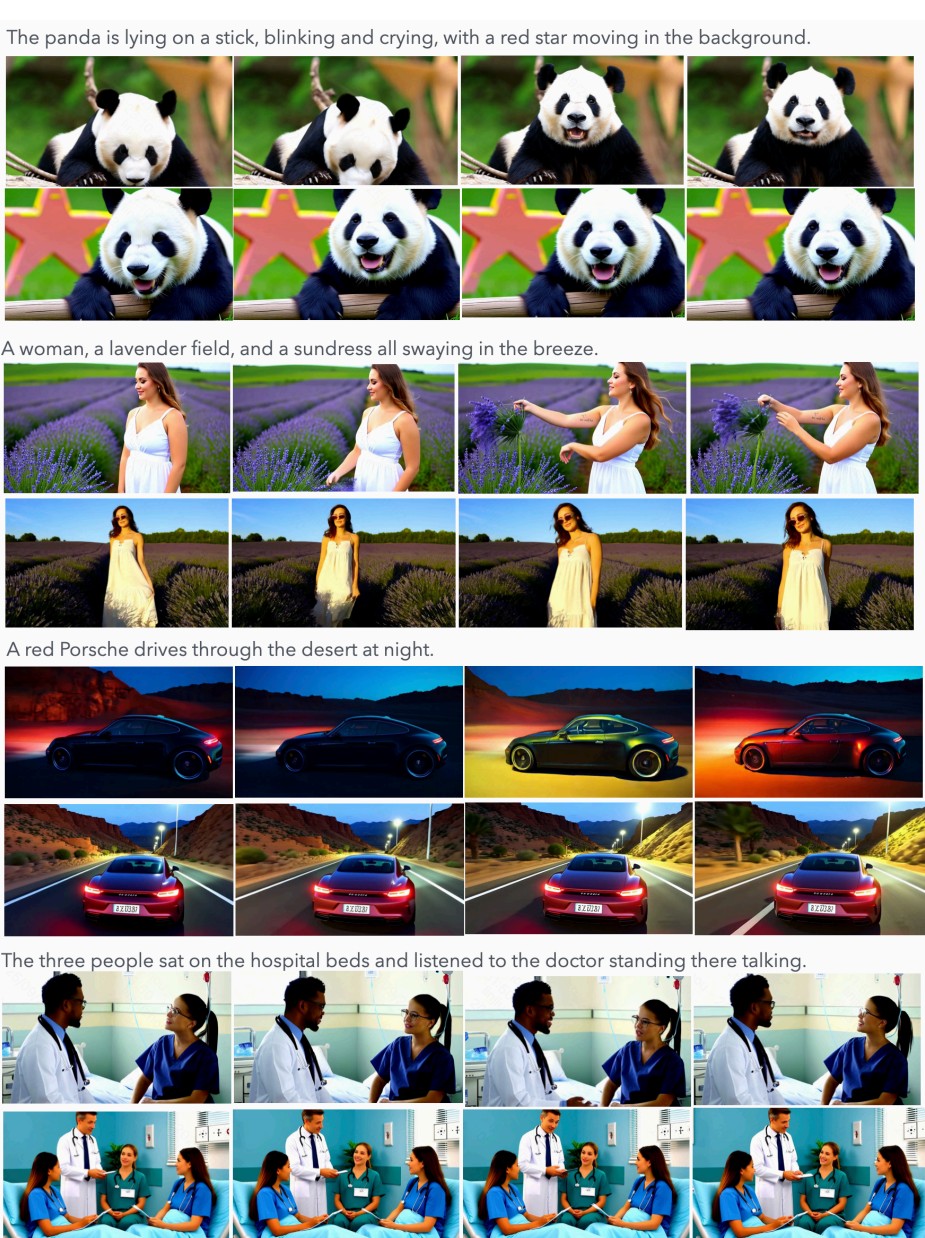

Figure 9: Visualization results of DM-Align (Group). From top to bottom, compared with the base model, our improvements include: correctly generating the red star, better character movement (the skirt flutters in the wind), correctly generating the red Porsche, and correctly generating three people.

