# OpenReview forum: "Unifying Reinforcement Learning and Distillation via Distribution Matching for Video Generation"
_ICLR.cc/2026/Conference — ICLR 2026 Conference Withdrawn Submission_

### Official Review · Reviewer_k1TU · 2025-10-31

**Soundness:** 3
**Presentation:** 2
**Contribution:** 3
**Rating:** 8
**Confidence:** 3

**Summary:**

This paper introduces a simple yet effective method for optimizing a T2V generative model from two perspectives: human preference alignment and inference efficiency. The proposed DM-PairLoss and DM-GroupLoss correspond to two popular RL methodologies—DPO and GRPO, respectively—both demonstrating effectiveness and yielding significant performance improvements.

**Strengths:**

- The proposed method is simple but effective. Inspired by DPO and GRPO, the authors introduce DM-PairLoss and DM-GroupLoss. When combined with the DMD loss, the approach simultaneously improves both the inference efficiency and the quality of a T2V generative model.
- Additional ablation/insight experiments (e.g., Fig. 5) support the contribution of each training loss.
- The final models achieve substantial performance improvements with only a few hundred training steps.

**Weaknesses:**

1. It will be helpful if authors can clarify the rationale for choosing five steps, provide sensitivity analyses (e.g., 1/3/5/10 steps), and maybe report the effectiveness on the training cost.
2. Since the VBench score is measured on a domain-specific, I wonder whether the proposed method will damage the generalization ability of the raw model.
3. Typo. error in Eq. 3 (parentheses)

**Questions:**

N/A

---

### Official Review · Reviewer_Tmq2 · 2025-10-31

**Soundness:** 2
**Presentation:** 2
**Contribution:** 2
**Rating:** 2
**Confidence:** 3

**Summary:**

This paper, *“Unifying Reinforcement Learning and Distillation via Distribution Matching for Video Generation”*, proposes a unified framework that jointly optimizes **alignment** and **distillation** within a single stage for video generation models. Building upon **Distribution Matching Distillation (DMD)**, the authors introduce two novel loss functions—**DM-PairLoss** (DPO-inspired) and **DM-GroupLoss** (GRPO-inspired)—that enable **direct human-preference alignment via distribution matching**. The method avoids reverse-SDE conversions and operates efficiently under few-step generation (e.g., 4-NFE). Applied to the Wan 2.1 text-to-video model, the unified framework shows consistent improvements over the base model, DMD2, Flow-DPO, and DanceGRPO on both **VBench** metrics and human preference evaluations.

In essence, the paper claims that coupling alignment and distillation through distribution matching yields **synergistic optimization**, improving both visual quality and human preference alignment.

**Strengths:**

1. **Unified optimization with practical value.** By merging RL-based alignment and DMD distillation into a single framework (Eq. 8, Fig. 2), the paper eliminates two-stage training and achieves efficient 4-step video generation without extra inference cost. This is a concrete engineering improvement over existing multi-stage pipelines;

2. The formulation of DM-PairLoss (Eq. 6) and DM-GroupLoss (Eq. 7) provides a simple yet effective way to embed human preference gradients into the DMD training loop—directly in the score-matching space, avoiding the need for explicit reward modeling or SDE back-conversion;

3. **Clear motivation and negative results discussion.** The authors openly analyze known failure modes (e.g., reward model giving high scores to noise and collapse after RL-after-distillation) and visualize training behavior (Appendix C–D), which adds transparency to their experimental reasoning

4. As reported in Table 1 and Table 2, both DM-Pair and DM-Group variants outperform strong baselines: average VBench scores reach 82.78 (Pair) and 84.40 (Group) at only 4 NFE, with substantial improvements in human preference (+48% / +30% over the base model). This demonstrates that the unified training leads to consistent gains across textual alignment, motion quality, and visual fidelity

**Weaknesses:**

1. **The derivation from the Bradley–Terry model to Eq. (6) is not rigorous or self-consistent.**
    - In Appendix A, the paper starts from the paired comparison likelihood under the Bradley–Terry model:
    $L_{\text{pair}} = -\log P_\theta(x^+ \succ x) = \log(1 + p_\theta(x)/p_\theta(x^+))$ (Eq. 10).
    Then, it assumes an imbalance $p_\theta(x^+) \gg p_\theta(x)$ and introduces a small parameter $\varepsilon = p_\theta(x)/p_\theta(x^+)$, yielding $\log(1+\varepsilon) \approx \varepsilon$ (Eq. 11).
    However, the next step—claiming equivalence (up to a constant) with $s_\theta(x) - s_\theta(x^+)$ (Eq. 12)—**changes the form of approximation**:
    $\log(1+\varepsilon) \approx \varepsilon$ is a first-order Taylor expansion, but $\log p_\theta(x) - \log p_\theta(x^+)$ is a logarithmic ratio, not a linear one.
    The substitution therefore mixes two incompatible approximations.
        - Moreover, when deriving the practical gradient, the paper transitions from Eq. (13) to Eq. (14) by treating the parameter gradient of $s_\theta(x^+)$ as if it could be propagated through $\partial x / \partial \theta$ along the sampling path. Yet $x^+$ is independent of θ (it is a fixed preferred sample), so this term should have **an explicit parameter derivative**, not a path derivative.
        - The resulting Eq. (6) (also Eq. 14 in the appendix) thus conflates parameter and path gradients, which makes the claimed connection between the Bradley–Terry loss and the DMD-style score difference **heuristic rather than formally justified**.
        These inconsistencies weaken the mathematical credibility of the proposed DM-Align loss and cast doubt on whether it actually optimizes the intended likelihood objective
2. **Using the fake model’s score for both preferred and generated samples introduces self-referential bias.**
    - In Eq. (6) and (7), the gradient is computed as the difference
    $s_{\text{fake}}(F(x^+, t),t) - s_{\text{fake}}(F(G_\theta(z), t),t)$.
    This means the *same fake model* estimates the scores for both $p_{\text{pref}}$ and $p_{\text{gen}}$.
    However, according to Algorithm 1 (lines 16–20), the fake model µ_fake is trained only with denoising supervision on generated data $x = G_\theta(z)$ (using `stopgrad(x)`), so it approximates $p_{\text{gen}}$ —not $p_{\text{pref}}$.
    Treating $s_{\text{fake}}(x^+)$ as an unbiased estimate of $\nabla_x\log p_{\text{pref}}(x^+)$ is thus **conceptually inconsistent**.
    In effect, the update direction becomes “move the generator toward regions that the current fake model already assigns high density to,” reinforcing existing generation modes rather than aligning toward human preferences.
    This can yield **mode attraction** or even self-collapse, where the model keeps amplifying its own biases instead of improving human-alignment fidelity;
3. **Discrepancy between the DM-GroupLoss description and its formal equation.**
    - The paper claims (lines 306–323) that low-advantage samples are shifted toward the averaged preferred output $\bar x^+$, whereas high-advantage samples remain closer to themselves.
    But the formal Eq. (7) simply multiplies a single global advantage A with the score difference $s_{\text{fake}}(F(\bar x^+,t),t) - s_{\text{fake}}(F(G_\theta(z),t),t)$.
    There is no conditional logic distinguishing low-A from high-A cases, nor any convex interpolation between $\bar x^+$ and $x$.
    This inconsistency between narrative and formula makes it unclear how the supposed adaptive behavior (“move more if low-advantage, less if high-advantage”) is actually implemented.
    Without that mechanism, Eq. (7) effectively reduces to a simple *globally scaled pairwise update*, which does not guarantee the intended gradient modulation across samples

**Questions:**

1. Could you present the exact mathematical sequence from Eq. (10) → (11) → (12) → (13) → (14), explicitly stating the assumptions under which each approximation holds (e.g., small-ε regime, independence of x⁺ and θ)? In particular, how is the parameter gradient of $s_\theta(x^+)$ replaced by a path-wise gradient through $G_\theta(z)$ when x⁺ does not depend on θ? A clear derivation or error bound would help justify Eq. (6).
2. What theoretical or empirical rationale supports approximating $\nabla_x\log p_{\text{pref}}(x)$ by $s_{\text{fake}}(x)$? Is there any analysis (e.g., bias–variance decomposition, consistency in the limit of perfect µ_fake) that guarantees this approximation does not introduce systematic bias? If not, would incorporating gradient information from the reward model or from a separate preference estimator mitigate this self-referential issue?
3. Please clarify whether Eq. (7) is implemented as a single global advantage-scaled update or as an adaptive convex combination between $\bar x^+$ and x for different A values. The text implies a two-regime behavior (“low-A → toward $\bar x^+$”; “high-A → toward x_i$”), but the equation does not reflect it.

---

### Official Review · Reviewer_qSsa · 2025-11-01

**Soundness:** 3
**Presentation:** 3
**Contribution:** 1
**Rating:** 2
**Confidence:** 3

**Summary:**

The paper introduces a distillation and alignment technique for flow matching models, combining techniques adopted from DMD and DPO/GRPO, and performs empirical studies on video generation tasks. Results show gains compared to baselines.

**Strengths:**

* The problem formulation and method are clearly presented.
* Empirical advantages over prior methods are validated in two dataset.

**Weaknesses:**

* In Figure 1., the paper claims that RL-after-distillation leads to model collapse, which seems to be verified in the "Flow-DPO + DMD2" and "DanceGRPO + DMD2". What about swapping the order, and run distillation-after-RL on the base model? Would the proposed method still show advantages?
* All experiments are run on Wan 1.3B model and the performance has not been verified on other flow model architectures. Results in this front would strengthen the paper.
* Main novelty that is explicitly discussed in the paper seems to be the DM-Align loss, which is a direct weighted summation of loss terms proposed in existing works (DMD, GRPO, DPO). The methodology contribution of the paper should be more extensively discussed.

**Questions:**

* Why using two different datasets for DM-Align (Pair)  and for DM-Align (Group)? Specifically, why not using ConsistID dataset for the DM-Align experiments (which shouldn't require ground truth labeling)?

---

### Note · Authors · 2025-11-13

**Comment:**

Thank you to all the reviewers for your thoughtful and constructive feedback. After careful consideration, we acknowledge that the current version of this work still has notable gaps in algorithm design, experimental methodology, and theoretical justification that prevent it from meeting the standards expected for publication.

We sincerely appreciate the time and effort each reviewer dedicated to evaluating our submission. In particular, we are deeply grateful to Reviewer Tmq2 for the exceptionally detailed and insightful comments—your rigorous critique has highlighted critical issues that will greatly inform our future revisions and research direction.

We plan to thoroughly address these concerns and significantly improve the work before considering resubmission elsewhere. Thank you again for your valuable input.

**Withdrawal Confirmation:**

I have read and agree with the venue's withdrawal policy on behalf of myself and my co-authors.